# Development of CDK4/6 Inhibitors: A Five Years Update

**DOI:** 10.3390/molecules26051488

**Published:** 2021-03-09

**Authors:** Alessandra Ammazzalorso, Mariangela Agamennone, Barbara De Filippis, Marialuigia Fantacuzzi

**Affiliations:** Unit of Medicinal Chemistry, Department of Pharmacy, “G. d’Annunzio” University, 66100 Chieti, Italy; alessandra.ammazzalorso@unich.it (A.A.); mariangela.agamennone@unich.it (M.A.); barbara.defilippis@unich.it (B.D.F.)

**Keywords:** cyclin-dependent kinase, cancer, resistance, small molecule inhibitors, PROTACs

## Abstract

The inhibition of cyclin dependent kinases 4 and 6 plays a role in aromatase inhibitor resistant metastatic breast cancer. Three dual CDK4/6 inhibitors have been approved for the breast cancer treatment that, in combination with the endocrine therapy, dramatically improved the survival outcomes both in first and later line settings. The developments of the last five years in the search for new selective CDK4/6 inhibitors with increased selectivity, treatment efficacy, and reduced adverse effects are reviewed, considering the small-molecule inhibitors and proteolysis-targeting chimeras (PROTACs) approaches, mainly pointing at structure-activity relationships, selectivity against different kinases and antiproliferative activity.

## 1. Introduction

Breast cancer (BC) is the most recurrent cancer in women worldwide, impacting 2.1 million women each year according to World Health Organization [1]. BC is a heterogeneous disease due to genetic factors that are reflected in different phenotypes. BC can be divided into different subtypes: luminal, in which estrogen receptors (ER) and/or progesterone receptors (PR) are expressed, further divided into luminal A and luminal B subtypes depending on the expression of Ki67 (low levels in A and high in B); HER2+, in which the human epidermal growth factor receptor 2 (HER2) is overexpressed and ER and PR are lacking; triple negative BC (TN), in which the previous targets are not expressed. The estrogen-receptor positive (ER+) BC is the most common type, with the prevalence of about 60% of cases in pre-menopausal women and 75% in post-menopausal women [2,3,4,5].

The anti-hormonal treatment involves the suppression or reduction of the estrogen effects and can be carried out using drugs that limit the production of these hormones, such as aromatase inhibitors (AIs), or act on the ER receptor, such as selective ER modulators (SERM) or down-regulators (SERD) [6,7,8]. The adjuvant therapy consists of 5–10 years of ER-directed endocrine therapy that result in a reduction of mortality in ER+ BC of more than 40%. Resistance to endocrine therapy leading to early-stage ER+ BC is common and decisive in the setting of advanced disease [9,10].

The aromatization reaction in the final step of estrogen biosynthesis is unique, therefore this reaction becomes an excellent target for inhibiting the synthesis of estrogens without affecting the production of other steroids. In recent decades, several aromatase inhibitors have been developed to adequately suppress estrogen production and have been used in the treatment of estrogen-dependent BC [11,12,13,14,15,16].

Given the high percentage of resistance to aromatase inhibitor treatment, new therapeutic strategies have been identified to make the treatment of ER+ BC more effective. One of the mechanisms involved in the resistance concerns the activation of cyclin-dependent kinases (CDKs) as an ER-independent growth signal, that involves the important protein kinase signaling pathway (PI3K/AKT/mTOR) (Figure 1) [17,18].

CDKs, a family of serine/threonine kinases, regulate cell cycle progression into the four distinct phases G1, S (DNA synthesis), G2 and M, and are crucially involved in the regulation of cell division and proliferation. CDK stability, activation and downstream phosphorylation is controlled by cyclin counterpart and endogenous inhibitors. To date, 21 CDKs are known and their role in different types of cancer has been reported by many research groups [19,20,21]. In particular, CDK1, 2 and 4 regulate the transition of the cell cycle steps, CDK 7, 8, 9 and 11 regulate the gene transcription, while CDK 6 regulates both [22,23,24].

Mitogenic, hormonal, and growth factors allow the cyclin D to bind CDK4/6, forming the complex that regulates the phosphorylation status of retinoblastoma protein (Rb). The phosphorylated-Rb determines the dissociation of the transcription factor E2F that binds to DNA and promotes the expression of different genes, regulates DNA replication and cell division, with the transition from G1 to S phase (Figure 1). Several endogenous factors control cell proliferation, including INK4 family proteins (p16, p15, p18, p19), cyclin inhibitory proteins and kinase inhibitory proteins (KIPs, p21, p27) [25,26].

The dysregulation of cell cycle caused by the overexpression or gain-of-function mutations in the CDKs and cyclins or the loss of endogenous inhibitors expression and function, is recurrent in many cancer diseases. In BC, but also in other type of cancer, a dysregulation of this process leads to a proliferative stimulus, and a significant role in this mechanism is played by the overexpression of cyclin D. Inhibition of cell cycle using CDK4/6 inhibitor has emerged as antitumor treatment in BC in association to the hormonal therapy to overpass resistance to AIs and avoiding relapses [27,28,29,30,31,32]. 

Over the years several CDK inhibitors have been developed and tested in different types of cancer [33,34,35,36,37]. First generation inhibitors (flavopiridol and roscovitine) demonstrated an inadequate balance between efficacy and toxicity, due to their action on several kinases (pan-inhibitors) [38,39,40]. Second generation inhibitors (dinaciclib) were developed with the aim to increase selectivity and potency, but demonstrated limited efficacy and considerable toxicity in clinical studies. The toxicity of these compounds is associated with the multi-target activity against isoforms fundamental for the proliferation (CDK1) and survival (CDK9) of normal cells [41,42,43,44]. However, in recent years, the interest in identifying inhibitors of specific kinases with targeted action on tumor cells and with less toxic effects, has led to the discovery of selective CDK4/6 inhibitors [30,45]. Third generation CDK inhibitors selectively inhibit CDK4/6 with potent efficacy and reduced toxicity, such as the FDA approved palbociclib (**1**), ribociclib (**2**), and abemaciclib (**3**) (Figure 2) [46,47,48,49]. 

The search for even more selective CDK4/6 inhibitors is still a challenge in the development of novel anticancer therapies. To date, three ATP-competitive CDK 4/6 selective inhibitors (trilaciclib, lerociclib and SHR-6390) are under advanced clinical trials. Trilaciclib (G1T28, **4**, Figure 3) is currently ongoing phase II trials in patients with small cell lung cancer (NCT02514447, NTC02499770) and with hormone receptor negative BC (NCT02978716), while entered in phase III for metastatic colorectal cancer (NCT04607668) [50,51,52,53]. Phase II (NCT02983071) trial of lerocliclib (G1T38, **5**, Figure 3) examined the effect in combination with fulvestrant in hormone receptor-positive, HER2-negative locally advanced metastatic BC while another phase II study combined lerociclib with osimertinib in EGFR-mutant non-small cell lung cancer (NCT03455829) [54,55]. SHR-6390 (**6**, Figure 3) is currently ongoing phase II trial in patients with hormone receptor positive, ErbB2 negative BC (NCT03966898) [56,57]. Moreover, an abemaciclib related compound, BPI-16350 (**7**, Figure 3), recently entered phase I clinical trial (NCT03791112) in patients with advanced solid tumor and the estimated study completion date is in December 2021 [58].

Based on their mode of action, kinase inhibitors can be divided into two types: ATP-competitive and non-competitive inhibitors. The reported CDK4/6 small molecule inhibitors abemaciclib, palbociclib and ribociclib are ATP-competitive inhibitors, forming hydrogen bonds in the ATP binding site with the kinase ‘‘hinge” residues (Val101 in CDK6) and hydrophobic interactions in the region normally occupied by the ATP adenine ring (Figure 4A,C,D) [59,60,61]. 

According to the study of Chen et al., the interactions enhancing the CDK inhibition and selectivity over other kinases are the H-bonds with the sidechains of His100 and Lys43, as shown for abemaciclib (Figure 4B) and a solvent-mediated interaction of a positively charged atom of the ligand and a solvent-exposed ridge consisting of Asp104 and Thr107 [62].

Given the great similarity in the structure of the approved compounds, several research groups have tried to identify different compounds with increased CDK4/6 selectivity, reduced adverse effects while maintaining or improving treatment efficacy. In this review we focused the attention on the last five years progress in the optimization of clinically approved CDK4/6 inhibitors, primarily tested for their BC anti-tumor activity. The main works, considering both the small-molecule inhibitors and the PROteolysis-TArgeting Chimeras (PROTACs) are summarized.

## 2. Small-Molecule Inhibitors

Small molecule inhibitors (SMIs) are compounds of ≤500 Da size, often administered orally, useful in cancer diseases. The traditional chemotherapy includes single or combination-therapy of drug targeting dividing tumor cells. The principal drawback of this non-targeted approach is the non-selective action on both normal and cancer cells. SMIs bind to specific molecular targets (targeted therapy) and selectively eliminate malignant cells [63,64]. The small size allows to use these molecules towards extracellular, surface, or intracellular proteins, including anti-apoptotic proteins that play a key role in cell growth and promotion of metastases. The most interesting targets in the antitumor field are mainly kinases such as serine/threonine/tyrosine kinases, matrix metalloproteinases (MMP), and CDKs [63,64,65,66,67,68].

The following SMIs are classified based on their chemical scaffold.

### 2.1. Thiazolyl-Pyrimidine Derivatives

Starting from the structure of abemaciclib and considering the fundamental interactions with the hinge region of CDK4/6, Tadesse et al. synthesized three series of compounds keeping constant pyrimidine, pyridine, and the amino linker to maintain the coplanarity of the two rings for ATP-mimetic kinase inhibitors (Figure 5), that represents the characteristic moiety of the three approved inhibitors [69]. 

The thiazole C2 amino moiety, introduced in previously synthesized CDK9 inhibitors, established a H-bond with the highly conserved Asp (Asp163 in the CDK6) residue among CDKs family of the Asp-Phe-Gly motif in the ATP-binding pocket, improving strong hydrophobic interaction of the methyl thiazole with the gatekeeper residues [70]. For this reason, the 2-amine-thiazole was introduced in C4 of the pyrimidine scaffold. The pharmacophore of the Tadesse’s lead compound is the *N*-(5-(piperidin-1-yl)pyridin-2-yl)-4-(thiazol-5-yl)pyrimidine (**8**–**10**, Figure 5). The main modifications of the first series (**8**, Figure 5) concerned the mono or di-substitution of the amine group (R_1_, R_2_) with methyl, ciclopentyl, phenyl or *iso*-propyl; the introduction of a fluorine atom in C5 of pyrimidine (R_4_) to optimize the pharmacokinetic properties; the N4 (R_5_) of the piperazine was substituted with a variety of ionizable groups. The most active compound (**8a**, Figure 5) contains a morpholine ring and a cyclopentyl substitution of the amine (K_i_ CDK4 = 0.004 μM, K_i_ CDK6 = 0.030 μM). Its antiproliferative activity was assessed by MTT assay in human leukemia Rb positive MV4-11 (GI_50_ = 0.209 μM), and in human breast cancer Rb negative MDA-MB-453 (GI_50_ = 3.683 μM) cell lines. The SAR analysis shows that the amino group in the thiazole ring could accept only a mono-substitution and the cyclopentyl is better than alkyl chain or aromatic ring; the introduction of the electron withdrawing trifluoromethyl increases the toxicity; the substitution of the second nitrogen atom of the piperazine with carbon and the exocyclic primary amino decreases activity and selectivity, while the introduction of oxygen increases the selectivity toward CDK4/6. 

In the second series of derivatives (**9**, Figure 5), the amino group on thiazole ring was replaced with alkyl, ether or thioether substituents, a cyano group or chlorine atom on C5 of pyrimidine was introduced, and the C4 of piperidine was replaced by an oxygen atom or secondary or tertiary amine substituted with alkyl or acetyl group [71]. Among the synthesized compounds, **9a** and **9b** (Figure 5) emerged with good CDK inhibition values (K_i_ CDK4 = 0.010 and 0.007 μM, K_i_ CDK6 = 1.67 and 0.042 μM, respectively) and antiproliferative activity in MV4–11 (GI_50_ = 0.591 μM and 0.456 μM, respectively). The antiproliferative activity of compounds **9a**–**b** was also evaluated in a panel of human cancer cell lines including breast, colon, ovary, pancreas, prostate, leukemia and melanoma. Noteworthy, the different antiproliferative activity in the MDA-MB-453 (GI_50_ = 3.32 μM and 4.17 μM, respectively) and the corrensponding Rb-deficient MDA-468 (GI_50_ = 8.03 μM and 7.16 μM, respectively) confirms the mechanism of action of these compounds which act in the presence of the intact Rb function. The potent effect on the growth of melanoma M249 (GI_50_ = 0.47 μM and 0.91 μM, respectively) and of resistant to dabrafenib M249R (GI_50_ = 0.27 μM and 0.91 μM, respectively) cell lines paves the way for a possible therapeutic use in melanoma. The SAR analysis highlighted that the presence of the nitrogen atom of the pyridine is fundamental for CDK4/6 selectivity, and the substitution of the amine group in C2 of the thiazole with alkyl, thioether or ether prevents the H-bond with the conserved Asp163, increasing the selectivity.

In another series of derivatives (**10**, Figure 5) the substitution of the pyridine with a benzene ring, the mono-substitution of the amino group of the thiazole, the introduction of fluorine atom or cyano group in C5 of pyrimidine, and the alkylation of the piperidine nitrogen or the introduction of morpholine or piperidine were studied [72]. Compound **10a** (Figure 5) was the most active in biological assays, with good CDK 4/6 inhibition (K_i_ CDK4 = 0.002 μM, K_i_ CDK6 = 0.279 μM), selectivity over other kinases and good pharmacokinetic profile. The antiproliferative activity was tested in different cancer cell lines such as leukemia or solid cancers (breast, colorectal, melanoma, ovarian, prostate).

### 2.2. Benzimidazolyl-Pyrimidine Derivatives

Zha et al. focused their attention to the substitution of the abemaciclib pyridine with a benzoimidazolyl in C4 and the introduction of tetrahydro-naphthyridine as substituent of the amino linker, with the aim to discover conformationally restricted analogs sharing improved activity and selectivity [73]. Compound **11** (Figure 6), discovered through the combination of structure-based drug design and traditional medicinal chemistry approaches, retains all key contacts between abemaciclib and CDK6, such as the edge-to-face interaction of the benzimidazole ring and Phe98 of the gatekeeper and the H-bonds of the amino pyrimidine with residues in the hinge loop. The tetrahydro-naphthyridine forms an additional water-mediated H-bond between the aromatic nitrogen with His100 and a salt-bridge interaction of the physiologically protonated nitrogen and Asp104 (Figure 4D) [19,74].

The authors explored the removal of the nitrogen in pyridine, in 2,4-pyrimidine or the substitution with a 4,6-pyrimidine, confirming that the nitrogen atoms are fundamental for CDK6 selectivity over CDK1. Although compound **11** exhibited good activity (IC_50_ CDK4 = 1.5 nM) and a selectivity index of 311, the very poor pharmacokinetic properties required further optimization. In compounds of the series **12** (Figure 6) a cyclic or acyclic alkyl substitution of the *iso*-propyl of the imidazole of compound **11** was introduced without a substantial improvement of activity, suggesting that the hydrophobic cleft of the protein could not host rigid or bulky groups. Compounds of the series **13** (Figure 6) contain a variety of substituents on the protonable nitrogen of tetrahydro-naphthyridine. The hydrophilic substitutions maintain the inhibition potency, with amide analogues suffering from poor exposure; the introduction of a *N*-alkyl piperidine improves the inhibitory activity and selectivity. In fact, compound **13a** (Figure 6) emerged as the best compound, with enzymatic CDK4 IC_50_ of 1.4 nM and selectivity CDK1/CDK4 around 850 (IC_50_ CDK1 = 1180 nM), good antiproliferative activity in Colo-205 cell line (IC_50_ = 0.057 μM), favourable *in vitro* metabolic properties (microsomal stability and CYP isoforms inhibition) and robust pharmacokinetic properties in mice and rats.

Wang et al. synthesized and tested a library of abemaciclib analogs [75]. The tetracycle scaffold (piperazine, pyridine, pyrimidine and benzimidazole) was kept constant, while small substituents were introduced on piperazine nitrogen (ethyl, 2-fluorethyl, cyclopropyl, *iso*-propyl), in C6 of pyridine (methyl), in C5 of pyrimidine (fluorine), and in C4 of benzimidazole (fluorine). The benzimidazole ring was transformed in a tricycle by connecting N1 and C2, inserting a cyclopentyl, cyclohexyl or cycloheptene [75]. 

A large library of 23 compounds (**14**, Figure 7) was tested for CDK1 and CDK4/6 activity. Compounds demonstrated null activity versus CDK1, but a remarkable inhibition of CDK4/6, with IC_50_ ranging from 0.6 to 340 nM. Compound **14a** (IC_50_ CDK4 = 7.4 nM and IC_50_ CDK6 = 0.9 nM) was further tested for hERG channel inhibition, showing low heart toxicity. The pharmacokinetic parameters of **14a** (Cmax, AUC, T_1/2_, MRT, CL/f, V2/F) demonstrated drug-like properties for following development. A docking study revealed that the di-methyl cyclopentyl group contributed to favourable H-bond between the nitrogen atom of the imidazole-condensed cycle and the amine group of Lys43.

The studies on Colo-205 subcutaneous xenografts tumor model in BALB/c nude mice revealed that compound **14a** was not well tolerated and had a narrow therapeutic window [73]. For this reason, Shi’s research group synthesized two novel series of benzimidazolyl-pyrimidine containing the tetrahydro-naphthyridines with a dimethylamino-ethyl group as substituent on protonable nitrogen of the bicycle: in the series **15** (Figure 8) the nitrogen atom of imidazole was substituted with alkyl or cycloalkyl, while in the series **16** (Figure 8) the protonable nitrogen of the ethylamine chain was substituted [76]. Compound **16a** (Figure 8) demonstrated nanomolar *in vitro* activity (IC_50_ CDK4 = 0.71 nM, IC_50_ CDK6 = 1.10 nM) with high kinase selectivity, excellent metabolic properties, good pharmacokinetic properties, low toxicity, and desirable antitumor efficacy in MCF-7, Colo-205, and A549 xenograft murine models. Even though compounds of series **16** possessed fair CDK4/6 activity in the range of nanomolar (IC_50_ = 0.999–6.14 nM), many of them failed in antiproliferative activity in MCF-7, T-47D, ZR-75-1, and Colo-205 cell lines.

The SAR analysis demonstrated that, with respect to *N*-*iso*-propyl (**16a**), the *N*-methyl or *N*-ethyl substitution of the imidazole decreased the activity more than others alkyl or cycloalkyl groups in terms of CDK 4/6 activity. The cycloalkyl, probably for its steric hindrance, also decreases the antiproliferative activities. Considering the substitution of *N*-methyl on the nitrogen of the ethylamine chain on tetrahydro-naphthyridine, the introduction of a bulkier group was unproductive in terms of CDK4/6 activity and selectivity over CDK2.

### 2.3. Pyrido-Pyrimidine Derivatives

Considering the pyrido [2,3-*d*]pyrimidine scaffold of palbociclib, Abbas et al. synthesized two series of 7-thienylpyrido[2,3-*d*]pyrimidines (**17** and **18**, Figure 9) and tested them for CDK6 inhibition and cytotoxicity against breast, lung, and prostate cancer cell lines [77]. In the series **17**, the 2-aryldiene hydrazinyl moiety was introduced in the scaffold and the aryl group was substituted in *para*-position. The most active compound of this series resulted **17a**, containing a *para*-methoxy group on the benzene ring. In fact, it demonstrated a CDK6 IC_50_ value of 115.38 nM and a good cytotoxicity against breast MCF-7 (IC_50_ = 1.59 μM), prostate PC-3 (IC_50_ = 0.01 μM), lung A-549 (IC_50_ = 2.48 μM) cancer cells.

The series **18** is constituted by a fused ring to pyrimidine forming a tricyclic pyridothyazolopyrimidine, substituted in C2 with a *para*-substituted benzylidene. Compound **18a** emerged as the most potent CDK6 inhibitor (IC_50_ = 726.25 nM) and cytotoxic against MCF-7 (IC_50_ = 0.01 μM), prostate PC-3 (IC_50_ = 1.37 μM), lung A-549 (IC_50_ = 1.69 μM) cancer cell lines.

### 2.4. Imidazo-Pyrido-Pyrimidine Derivatives

The pyrido-pyrimidine scaffold of palbociclib was fused with imidazole in two novel series of CDK4/6 inhibitors containing the fused tricyclic ring of imidazo[10,2′:1,6]pyrido[2,3-d]pyrimidine (**19** and **20**, Figure 10) [78].

In the series **19**, the fused tri-heteroaryl structure was substituted with a methyl in C5 and C8, cyano in C6, while the amino group in C2 was substituted by phenyl or para-substituted phenyl groups. This type of modification did not sufficiently improve the activity, and compound **19a** with the piperazine in *para* position showed modest inhibition values (IC_50_ CDK4 = 26.50 nM, IC_50_ CDK6 = 33.60 nM). Keeping constant the piperazine moiety, in the series **20** the C6 and C8 positions were changed by the introduction of methyl, *iso*-propyl, *terz*-butyl, cyclopentyl, cyclohexyl, phenyl, ethylester, or pyrrolidine-1-carbonyl in C8, while the cyano group in C6 was replaced with the acetyl one. Compound **20a** was the best one of the series in terms of inhibition (CDK4 IC_50_ = 0.8 nM, CDK6 IC_50_ = 2.0 nM).

The piperazine ring of compound **20a** was also replaced by saturated heterocycle, distanced by a methyl and a carbonyl linker; alternatively the piperazinyl-pyridine portion was replaced with a fused bicycle. None of these changes improved the inhibition of kinases [79]. Compound **20a** demonstrated good activities on Colo-205 (IC_50_ = 56.4 nM), and glioma U87MG (IC_50_ = 84.6 nM) cell lines, favourable *in vitro* metabolic properties (microsomal stability, CYP isoforms inhibition), acceptable pharmacokinetic profiles in mice and rats, antitumor efficacy with controllable observed side effects in xenograft in vivo studies.

### 2.5. Pyrazolo-Quinazoline Derivatives

Considering the inhibition activity of different kinases (Aurora-A, CDK2, Polo-like Kinase 1) of the 4,4-dimethyl-4,5-dihydro-1*H*-pyrazolo[4,3-h]quinazoline [80], Zhao et al. synthesized a series of 4,5-dihydro-1*H*-pyrazolo[4,3-h]quinazolines (**21**, Figure 11) and tested their inhibition of CDK4/6 [79]. The amine group in C2 position of the quinazoline was substituted with pyridine or benzene ring. Compounds containing the pyridine confirmed that the nitrogen atom in this position affects not only the inhibitory activity, but also the cellular activity against MCF-7 cell line. In fact, the pyridine derivatives were more active as CDK4/6 inhibitors and displayed improved cellular activity.

Compound **21a** was the best one of this series, showing good activity on CDK4/6 (IC_50_ CDK4 = 0.01 μM, IC_50_ CDK6 = 0.026 μM) and high selectivity against CDK2 (IC_50_ CDK2 = 0.70 μM), anti-proliferative activity in MCF-7 cell line (IC_50_ = 0.19 μM) and other solid tumors (colorectal, liver, pancreatic), favorable pharmacokinetic parameters (T_1/2_, CL, AUC, V, Cmax).

## 3. PROTACS

The therapeutic use of small-molecule inhibitors to target proteins such as transcription factors, non-enzymatic, and scaffolding proteins, has several limitations because these targets lack appropriate active site to be occupied that directly modulate protein functions [81]. Moreover, high systemic drug exposures in the use of small molecules that bind to the active site of a protein are required to achieve site occupation, which may lead to an increase in adverse effects caused by binding to off-target sites [82]. Other complications in the prolonged use of small molecule inhibitors are the possible mutation of the target protein and the establishment of resistance to the therapy, the overexpression of such protein to balance the inhibition drug-mediated, and the accumulation. These mechanisms are associated with the partial or overall suppression of the downstream signaling pathways [83].

A strategy to circumvent the problem of binding site occupancy to regulate the inhibition of a protein and the possibility of significantly expand the number of proteins that can be inhibited, is the use of small-molecule-induced protein degradation. In this way, the pharmaceutical advantages deriving from the use of small molecules are preserved and the proteins generally considered “undraggable” are removed [84]. These hybrid molecules, generally called PROteolysis-TArgeting Chimeras (PROTACs), are constituted by two small binding molecules connected by a linker (Figure 12): one domain is directed to the targeted protein, while the other domain binds E3 ubiquitin ligase. The complex allows the binding of the proteolytic ubiquitin on the target protein, and its consequent degradation of the targeted protein in proteasome. PROTACs act catalytically and are not destroyed as small molecule suicide inhibitors that permanently bind target macromolecules [85,86].

PROTAC strategy is widely applied to degrade proteins related to immune disorders, neurodegenerative diseases, viral infections, and cancer diseases [87,88,89]. In this paragraph the application of PROTAC strategy to CDK4/6 inhibitors is summarized.

The use of this approach could be exploited to selectively inhibit CDK6 with respect to CDK4, which have specific functions, could derive from the use of PROTACs. In fact, the binding site of ATP in kinases 4 and 6 possesses a high structural similarity, that could hardly be circumvented with the use of small molecules. All the reviewed studies have in common the binding of the E3-binding portion (E3 ligase ligand) to the nitrogen atom of the piperazine of the three approved CDK4/6 inhibitors. In fact, the crystallographic studies of palbociclib, ribociclib and abemaciclib show that the piperazine ring is projected towards the solvent (Figure 4B), in an optimal position to act as an anchor point.

Among the first studies reporting a PROTAC active towards CDK4/6, emerges the work of Zhao and Burgess [90], who combined palbocilib and ribocilib with pomalidomide (cereblon (CRBN), E3 ligase ligand) by means of a linker containing a triazole ring (**22a**–**b**, Figure 13). Studies on MDA-MB-231, a triple negative breast cancer cell line, showed that CDK4 is degraded more efficiently and PROTAC containing palbociclib (**22a**) is more potent (DC_50_ CDK4 = 12.9 nM, DC_50_ CDK6 = 34.1 nM) than **22b** (DC_50_ CDK4 = 97 nM, DC_50_ CDK6 = 300 nM). The same CDK degradation and cytotoxicity studies conducted on MCF-7 showed that **22a**–**b** are less efficient towards this cell line with respect to the triple negative cell line.

In the same period, Rana et al. synthesized a chimera series of palbociclib and pomalidomide by changing the length and the composition of the flexible linker (**23**, Figure 14) [91].

All compounds with shorter linker degrade CDK6 partly, while the PROTAC containing the longest linker (**23a**, Figure 14) selectively degraded CDK6 at the single dose of 500 nM in pancreatic cancer MiaPaCa2 cells with respect to other cyclin-dependent kinases including CDK4. Two hypotheses on the selective behavior of this PROTAC could be found in the less stable ternary complex palbociclib-E3 ligase-CDK4, that avoids the degradation, or in the fast deubiquitination of CDK4. The quantitative degradation of only CDK6 (CDK4 was not affected) was observed for compound **23a** in a dose-response study at 4 and 24 h in Human Pancreatic Nestin-Expressing ductal (HPNE) and MiaPaCa2 cells at 100 nM.

Another library of PROTACs containing CDK4/6 inhibitor and pomalidomide was synthesized by Su et al. (**24**, Figure 15), in which the influence of the length and rigidity of the linker, the spatial orientation of the target protein and the E3 ligase, and the binding affinity of PROTAC to CDK4 and 6 were studied [92].

PROTACs containing ribociclib did not degrade CDK6, while for the others best results in selectively degradation CDK6 was obtained with shorter linkers. In particular, the linker anchoring group to CDK inhibitors (amide, triazole, or methylene) did not influence the activity while to the other side, the best anchoring group to E3 ligase was the amino group, demonstrating that the flexibility of this portion is fundamental to correctly interact. The most potent PROTAC **24a** possesses a DC_50_ value of 2.1 nM in glioblastoma U251 cells and demonstrated good potency also in hematopoietic cancer cells, including multiple myeloma MM.1S (IC_50_ 10 nM).

Jiang and co-workers prepared a library of palbociclib, ribociclib, and abemaciclib PROTACs (**25**–**27**, Figure 16) connected to pomalidomide through an alkyl or polyethylene glycol (PEG) linker [93].

PROTACs of each CDK4/6 inhibitor demonstrated degrading activity of both CDK4 and 6, but abemaciclib-PROTACs also induced the degradation of the off-target CDK9, that should be avoided [60]. The type of the linker (length and structure) and the CDK4/6 inhibitor of the PROTAC influenced the selectivity of degradation at 100 nM: compound **25a** (alkyl linker conjugated to palbociclib) indifferently degraded both CDK4 and CDK6, **25b** (extended PEG-3 linker conjugated to palbociclib) selectively hit CDK6, while **26a** (4-carbon alkyl linker conjugated to ribociclib) was selectively toward CDK4.

Compounds containing the imide group were tested for their capability to inhibit Ikaros (IKZF1) and Aiolos (IKZF3), well-established targets of imide-based degraders [94,95,96]. Compounds **25a**–**b** and **26a** degraded also IKZF1/3, resulting in an enhnanced anti-proliferative effect on mantle cell lymphoma lines.

Compound **25c** was previously synthesized by Brand and co-workers and studied for its ability to selectively degrade CDK6 over CDK4, in particular the correlation between the use of the CDK6 degrader in acute myeloid leukemia cells was investigated [97].

In a recent study, Anderson et al. evaluated the effect of other E3 ligase, such as von Hippel-Lindau (VHL) and Inhibitor of Apoptosis (IAP) instead of CRBN, on the selective degradation of CDK4/6, maintaining the anchoring on the nitrogen of piperazine of palbociclib and using different types of linkers (**28**, Figure 17) [98].

The dose-response study in Jurkat cells after 24 h revealed that the degradation of CDK4 and CDK6 occurred independently of the type of E3 ligases (VHL, CRBN, and IAP binder), with a CDK4 pDC_50_ in the range of 6.2–8.0 and CDK6 pDC_50_ in the range of 7.7–9.1. It is important to note that all of them show a greater degradation power towards CDK6, probably due to a better stability of the formed ternary complex.

Compounds **28a**–**b**, containing VHL and IAP, are the less potent degraders (**28a**: pDC_50_ CDK4 = 5.6; pDC_50_ CDK6 = 5.3; **28b**: pDC_50_ CDK4 = 6.7; pDC_50_ CDK6 = 5.8), probably due to the linker nature. The most potent CDK4/6 degrader is **25a**, previously reported by Jiang (pDC_50_ CDK4 = 8.0, pDC_50_ CDK6 = 9.1) [93].

Compounds **25a** was taken into account by Steinbach et al. to synthesize novel palbociclib based PROTACs by changing the E3 ligase portion and inserting various linkers (**29**–**31**, Figure 18) [99]. In the series **29**, pomalidomide was linked by an amide linker to the palbociclib piperidine, avoiding the protonation of the previously synthesized tertiary amine that could affect activity and selectivity. The linkers were polyethylene or alkyl chain of different size. These compounds were tested in multiple myeloma (MM.1S) cell lines at 0.1 μM and the activity of PROTACs was shown as the percentage of remaining CDK levels (D). The degradation percentage (D) of CDK6 for compounds **29a**–**c** was in the range of 7.7–8.4 and the selectivity over CDK4 in the range of 1.9–3.3. In the series **30** and **31**, palbociclib was linked to VHL ligand functionalized in two different positions to create an amide or a phenoxy group in the E3 ligase ligand side, while in the other side of the linker there was an amine group. Compound **30a** showed a degradation percentage 1.7 and a selectivity CDK4 ratio of 19, while compound **31a** showed a comparable degradation activity (D CDK6 = 1.4) but an improved selectivity (D_CDK4_/D_CDK6_ = 31). PROTACs **30a** and **31a** were also tested in different cancer cell lines (multiple mieloma, acute myeloid leukemia, acute lymphoid blastic leukemia) inhibiting cell proliferation.

## 4. Conclusions

Since 2015, the arsenal of drug against breast cancer is enriched with third-generation CDK4/6 inhibitors. Three compounds (palbociclib, ribociclib, abemaciclib) have been approved by the FDA for the treatment of breast cancer in association with endocrine therapy. These ATP-competitive compounds share a common portion interacting with the ATP-binding site; in fact, they contain the pyridine-amine-pyrimidine scaffold, that determines the formation of more than one H-bond with the hinge residue of the target kinases.

In the last five years, a number of small molecule inhibitors have been synthesized and tested in order to identify compounds more potent, selective, and with improved pharmacokinetic parameters. The main heteroaromatic scaffold, that represents the central part of the molecule, was kept constant, while different groups or additional cycles were introduced on the terminal portions.

The use of PROTACs (proteolysis targeting chimeras), composed combining the CDK4/6 inhibitor small molecule and an E3 ligase ligand, is a novel approach to selectively degrade the targeted kinases. The anchoring point in CDK inhibitor is the nitrogen of the piperazine, which is extended towards the outside of the binding site, without interfering with the ATP-binding site. The majority of studies have been done on palbociclib and pomalidomide, by varying the type (nature and length) of the linker, although studies on the other two approved CDK inhibitors and different E3 ligases are reported. These studies have shown that it is possible to selectively degrade CDK4 or CDK6, depending on the type of inhibitor and linker, although the single inhibitor acts to a comparable extent on the two kinases.

Moreover, in addition to the study on breast cancer, the actions on other cancer cell lines have been explored. The development of new CDK inhibitors or degraders will certainly continue over the next years and possibly will allow to treat other forms of cancer with improved potency and less side effects.

## Figures and Tables

**Figure 1 molecules-26-01488-f001:**
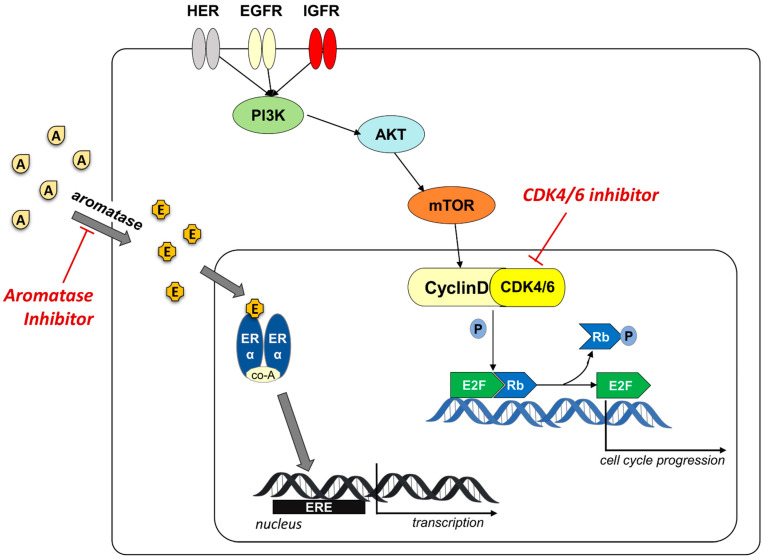
Schematic representation of the cyclinD/CDK4/6 involvement in overpassing resistance to aromatase inhibitors. Aromatase converts androgens (A) in estrogens (E) that bind to ER receptor. The recruitment of co-activators (co-A) allows the binding to ERE element on the target genes and the activation of the transcription. AIs block the production of estrogen inhibiting the ER-driven activation of cell cycle progression. The activation of cyclinD/CDK4/6 complex, mediated by the protein kinase signaling pathway (PI3K/AKT/mTOR), stimulates cell proliferation independently from aromatase. The use of CDK4/6 inhibitor blocks this alternative activation pathway.

**Figure 2 molecules-26-01488-f002:**
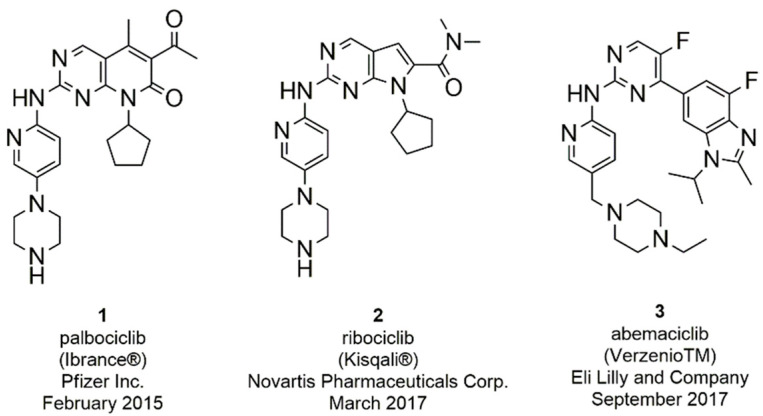
FDA approved CDK4/6 inhibitors palbociclib, ribociclib, and abemaciclib.

**Figure 3 molecules-26-01488-f003:**
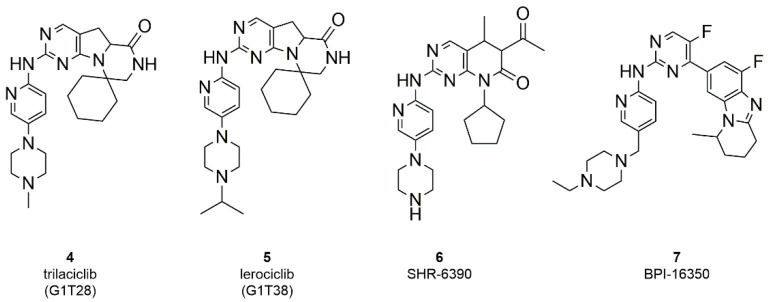
Chemical structure of ATP-competitive CDK 4/6 selective inhibitors in clinical trials.

**Figure 4 molecules-26-01488-f004:**
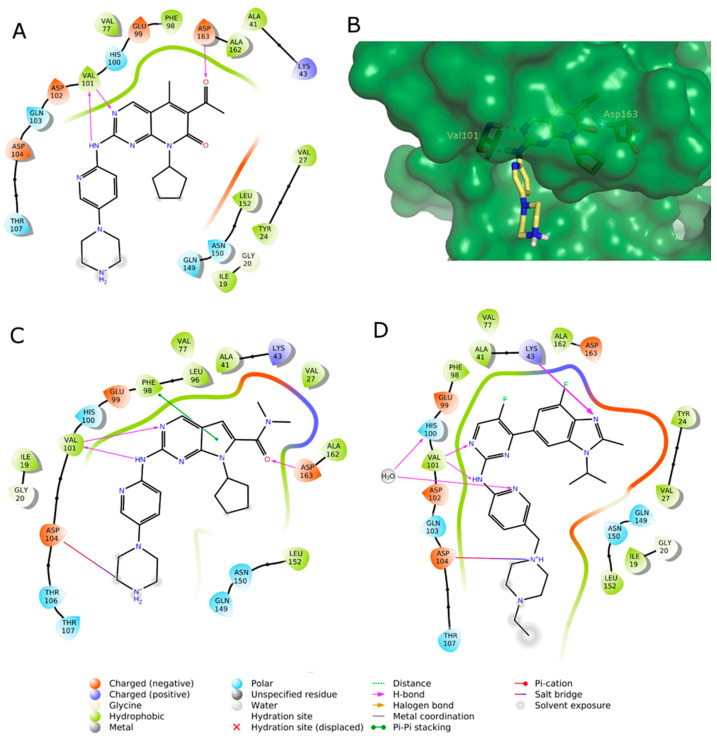
(**A**,**C**,**D**) 2D interaction diagram of the three approved CDK4/6 selective inhibitors as observed in their X-ray complexes into CDK6: (**A**) palbociclib (PDB ID: 5L2I); (**C**) ribociclib (PDB ID: 5L2T); (**D**) abemaciclib (PDB ID: 5L2S). (**B**) 3D representation of the X-ray binding geometry of palbociclib (stick, yellow C atoms) into the CDK6 binding site (dark green solid surface). Protein residues involved in key H-bond interactions are represented as stick; H-bonds are depicted as magenta dashed lines.

**Figure 5 molecules-26-01488-f005:**
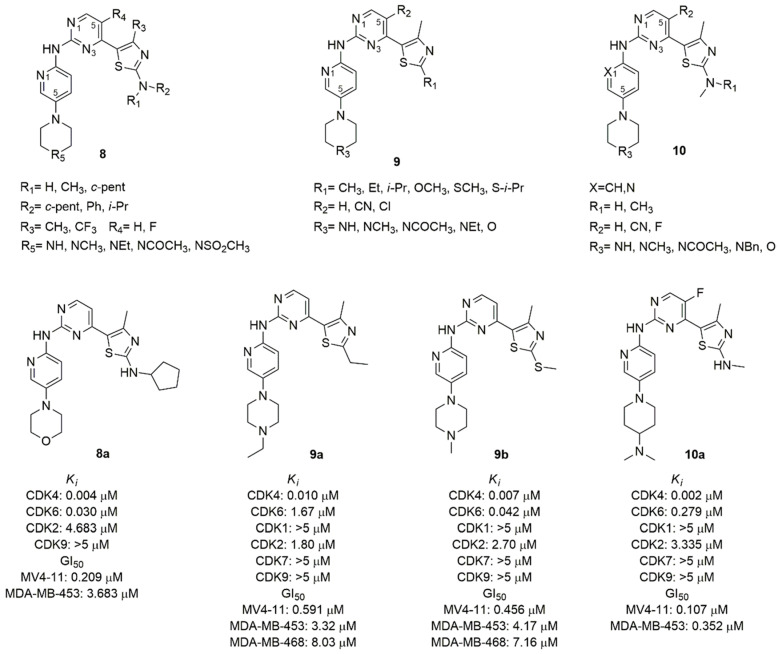
CDK4/6 inhibitors based on thiazolyl-pyrimidine scaffold.

**Figure 6 molecules-26-01488-f006:**
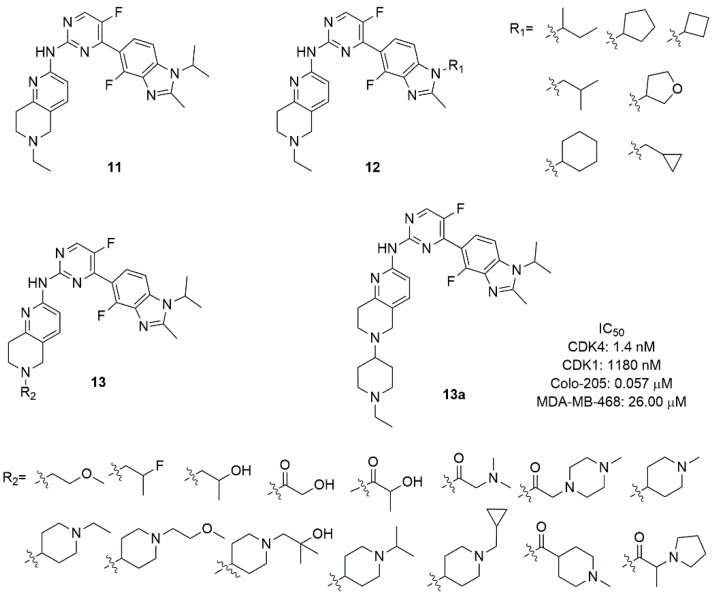
CDK4/6 inhibitors based on benzimidazolyl-pyrimidine scaffold.

**Figure 7 molecules-26-01488-f007:**
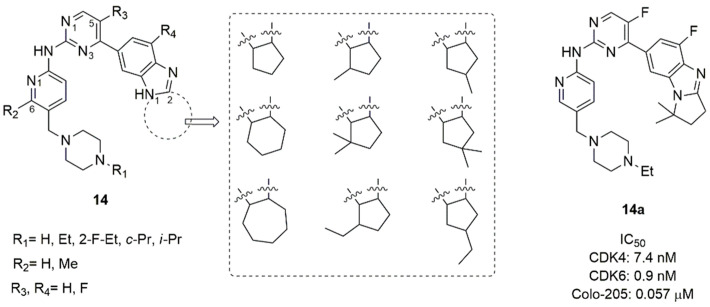
CDK4/6 inhibitors based on benzimidazolyl-cycloalkyl-pyrimidine scaffold.

**Figure 8 molecules-26-01488-f008:**
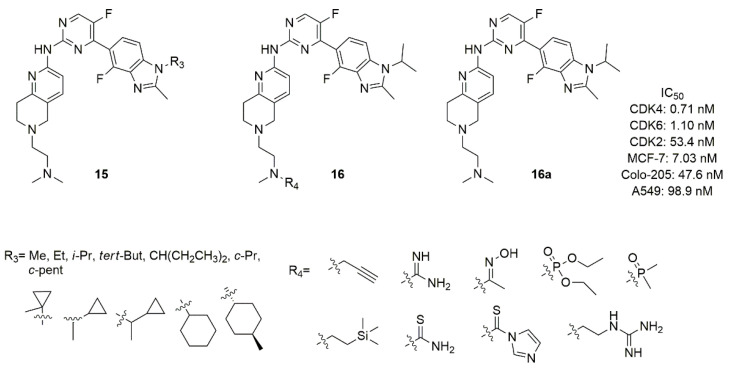
Structure of Shi’s research group of benzimidazolyl-pyrimidine derivatives.

**Figure 9 molecules-26-01488-f009:**
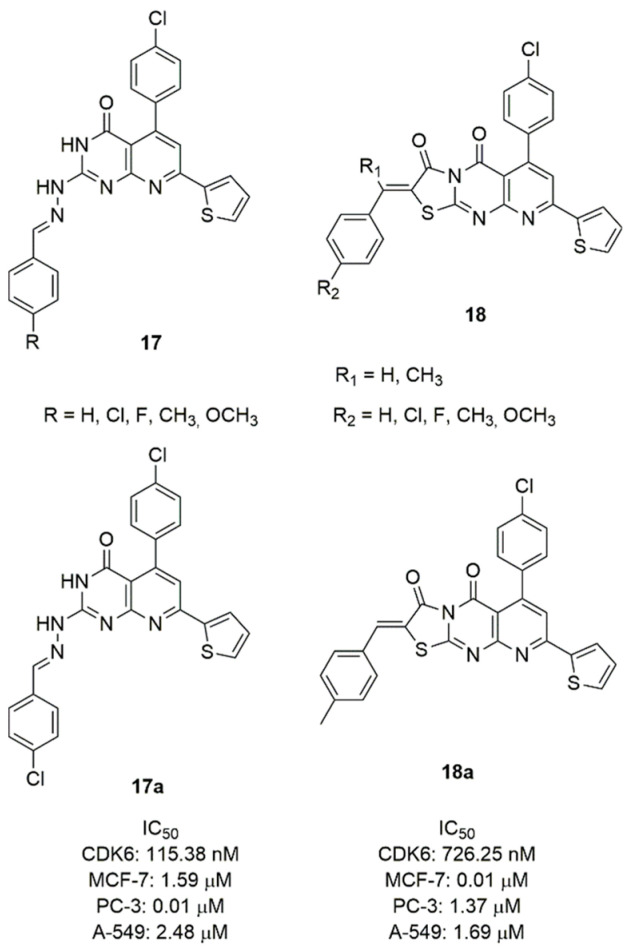
CDK6 inhibitors based on the pyrido-pyrimidine scaffold.

**Figure 10 molecules-26-01488-f010:**
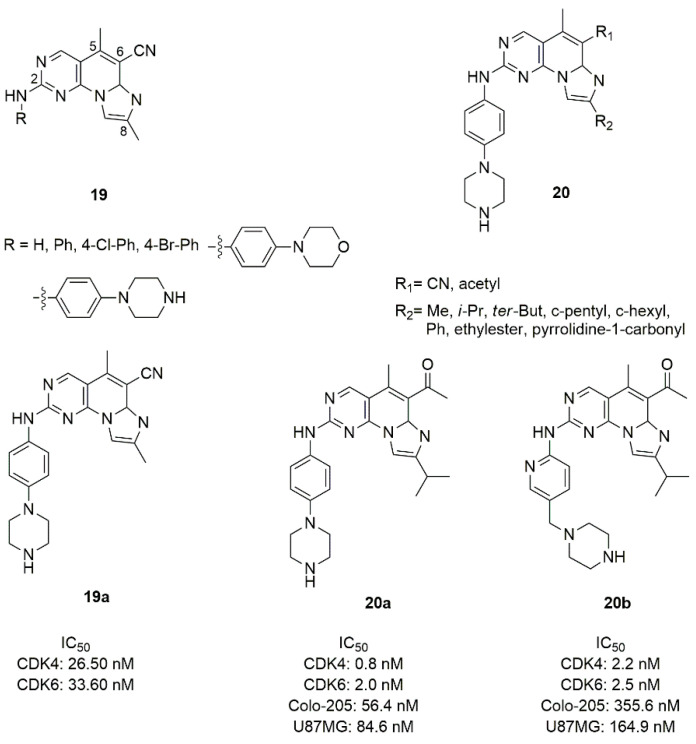
Structure of imidazo[10,2′:1,6]pyrido[2,3-d]pyrimidine derivatives.

**Figure 11 molecules-26-01488-f011:**
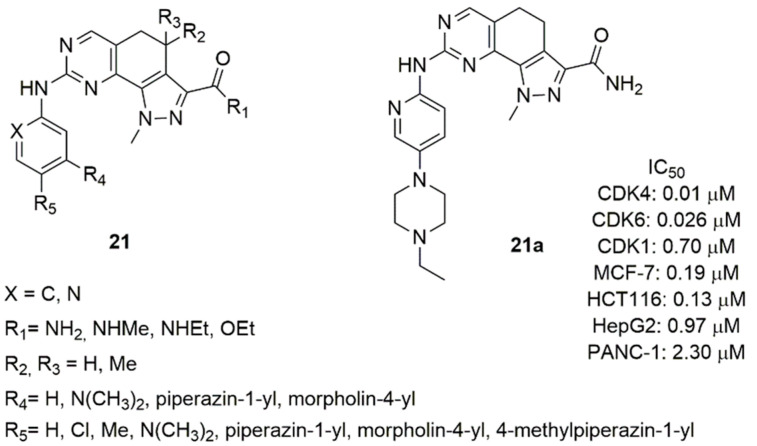
Structure of 4,5-dihydro-1H-pyrazolo[4,3-h]quinazoline derivatives.

**Figure 12 molecules-26-01488-f012:**
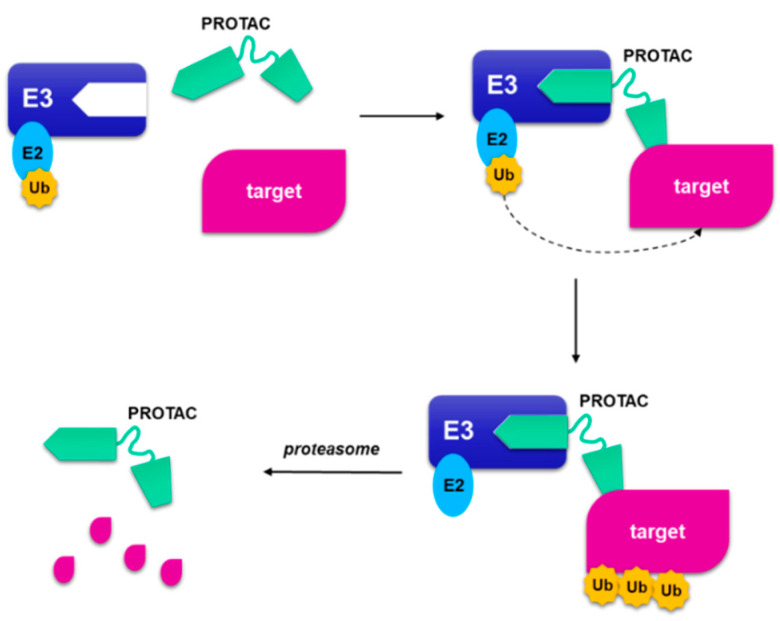
A schematic representation of proteolysis targeting chimera. The PROteolysis-TArgeting Chimera (PROTAC) is composed by a portion that binds to the ubiquitin ligase and a small molecule that binds to target protein, joined by a linker. When the targeted protein binds the small molecule, and the other part binds to E3 ligase, a ternary complex is formed. The following poliubiquitination of the target allows the proteasome to degrade the target protein and regenerate the PROTAC.

**Figure 13 molecules-26-01488-f013:**
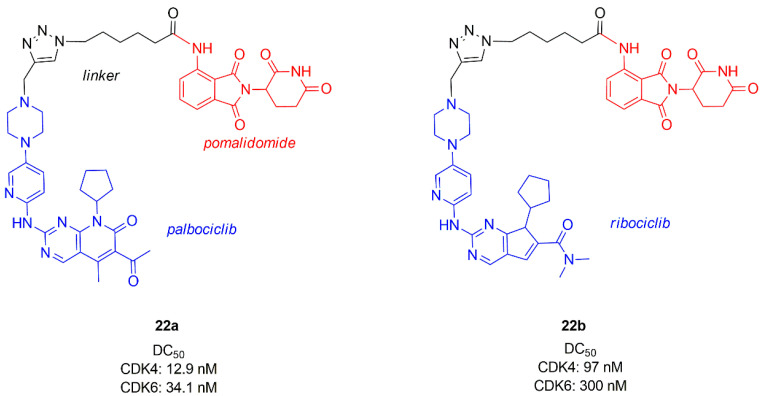
Chemical structures of palbociclib or ribociclib/pomalidomide PROTACs.

**Figure 14 molecules-26-01488-f014:**
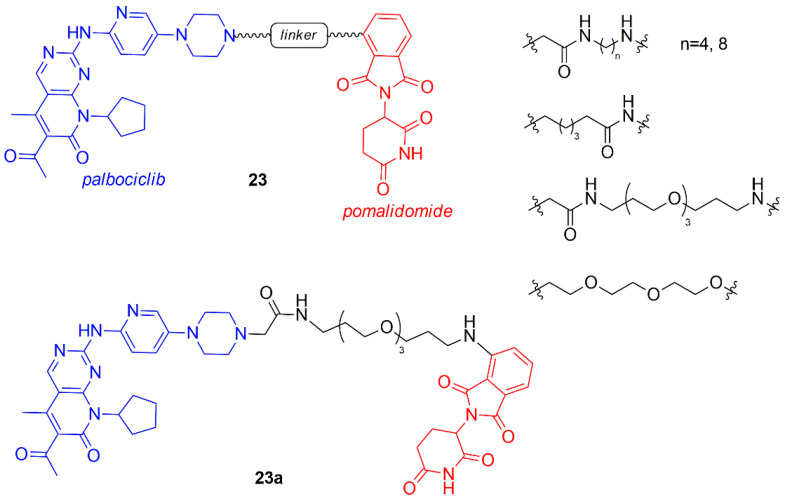
Chemical structures of palbociclib/pomalidomide PROTACs.

**Figure 15 molecules-26-01488-f015:**
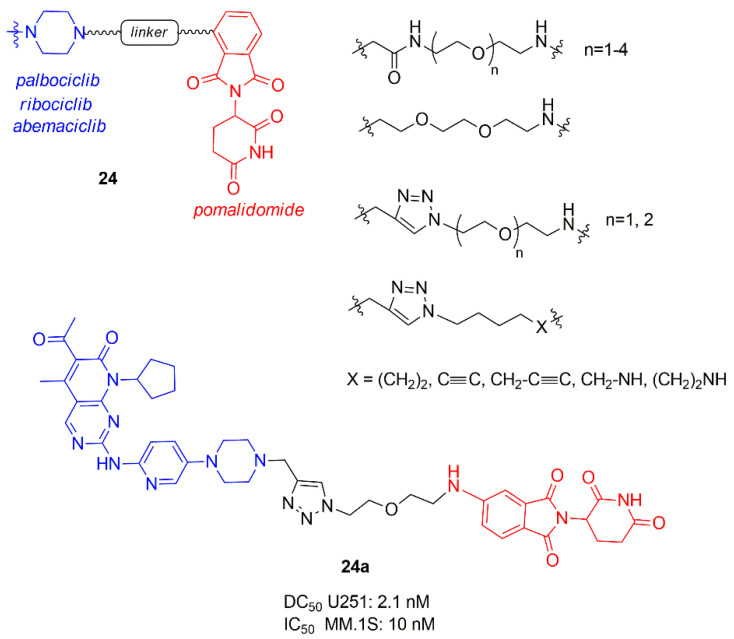
PROTACs containing CDK4/6 inhibitors and pomalidomide synthesized by Su et al.

**Figure 16 molecules-26-01488-f016:**
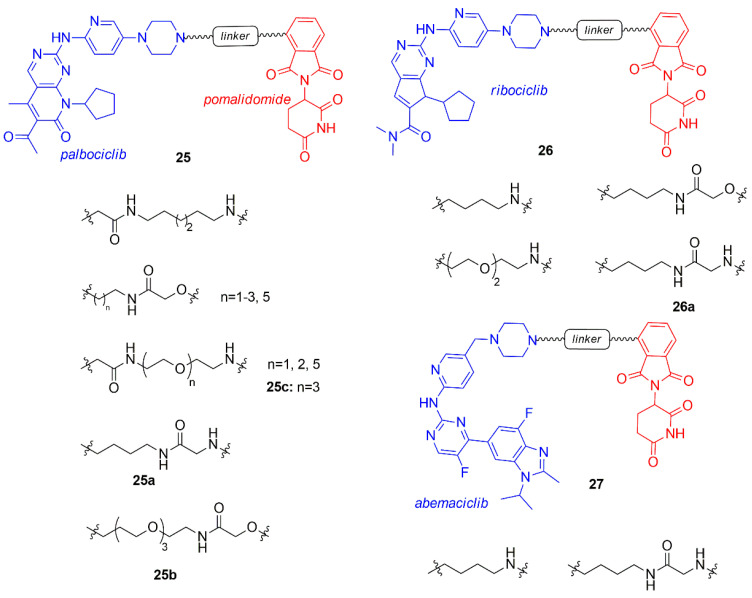
PROTACs containing CDK4/6 inhibitors and pomalidomide synthesized by Jiang et al.

**Figure 17 molecules-26-01488-f017:**
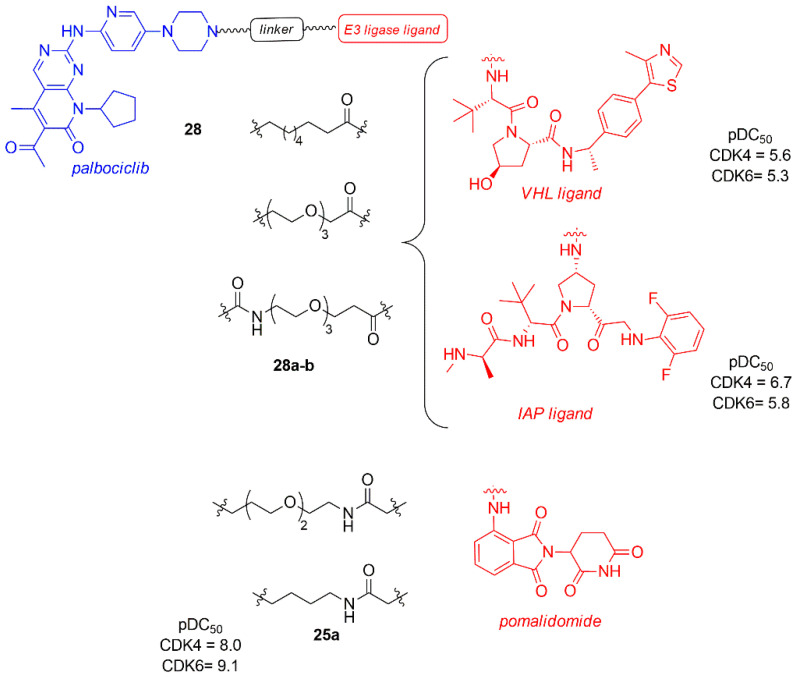
PROTACs palbociclib and E3 ligase ligands, such as von Hippel-Lindau (VHL) and Inhibitor of Apoptosis (IAP) ligands and pomalidomide.

**Figure 18 molecules-26-01488-f018:**
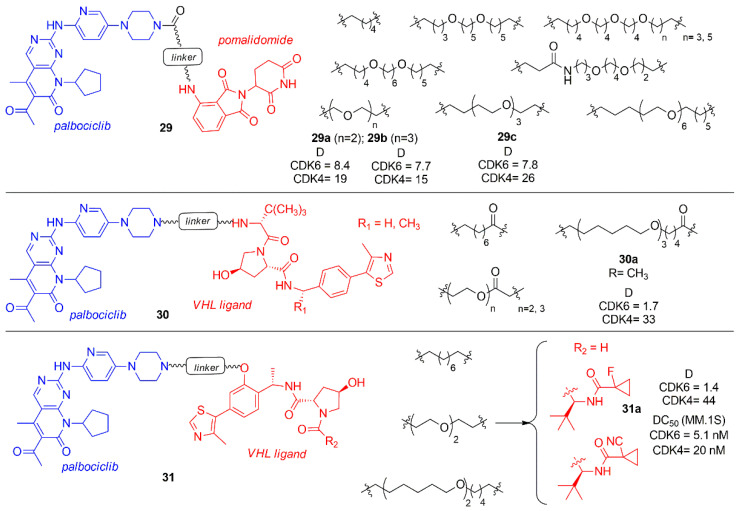
PROTACs containing palbociclib and E3 ligase ligand, such as pomalidomide and VHL ligand.

## Data Availability

The data presented in this study are available in this article.

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
