# Peer review of "Development of CDK4/6 Inhibitors: A Five Years Update"

_molecules, 2021, doi:10.3390/molecules26051488_

Round 1

Reviewer 1 Report

The manuscript by Alessandra Ammazzalorso etc. reviewed the recent updates on the development of CDK4/6 inhibitors. Generally, this review was well organized, which could be accepted for publication after addressing the following questions.

1. The recent developments summarized in the current form of this review mainly focused on abemaciclib, more developments (SAR studies) about palbociclib and ribociclib should be supplemented.

2. In the conclusions part, the authors concluded that "in addition to the study on breast cancer, the action on other cancer cell lines, such as acute myeloid leukemia, have been explored." To support this, more detailed preclinical and clinical progress about CDK4/6 inhibitor in the treatment of other cancers should be added in the revised form of this review.

3. The key CDK inhibitory potency, selectivity and cellular activity of the representative compounds should be listed following the chemical structures in the figures.

4. The language should be polished and the spelling mistakes should be corrected. For example, in line 26, HER2 should be the abbreviation of human epidermal growth receptor 2 instead of human epidermal growth factor 2.

Author Response

The manuscript by Alessandra Ammazzalorso etc. reviewed the recent updates on the development of CDK4/6 inhibitors. Generally, this review was well organized, which could be accepted for publication after addressing the following questions.

We would like to express our appreciation to the reviewer for suggesting how to improve the quality of the manuscript.

  1. The recent developments summarized in the current form of this review mainly focused on abemaciclib, more developments (SAR studies) about palbociclib and ribociclib should be supplemented.

Following the advice of referees, Section 2 has been organized into sub-sections, according to the chemical structure of the compounds. The derivatives of palbociclib are now reported in paragraphs 2.3- 2.5. Ribociclib analogues (synthesized in the last five years) are not mentioned in this paper as they were tested only in pancreatic cancer cells. (Shi, X. et al., Bioorg. Med. Chem. Lett. 33 (2021) 127725, doi: https://doi.org/10.1016/j.bmcl.2020.127725)

  1. In the conclusions part, the authors concluded that "in addition to the study on breast cancer, the action on other cancer cell lines, such as acute myeloid leukemia, have been explored." To support this, more detailed preclinical and clinical progress about CDK4/6 inhibitor in the treatment of other cancers should be added in the revised form of this review.

Thank you for your valuable suggestion. In this revised version, CDK4/6 inhibitors ongoing clinical trials were presented in the introduction section. Furthermore, the IC50 values in different cancer cell lines were entered for the most interesting compounds.

  1. The key CDK inhibitory potency, selectivity and cellular activity of the representative compounds should be listed following the chemical structures in the figures.

According to reviewer’s advise, inhibitory potency and cellular activity of representative compounds have been introduced in the figures.

  1. The language should be polished and the spelling mistakes should be corrected. For example, in line 26, HER2 should be the abbreviation of human epidermal growth receptor 2 instead of human epidermal growth factor 2.

The language was carefully revised, and spelling mistakes were corrected.

Reviewer 2 Report

Reviewer’s Comments to Author 

Current manuscript describes recent updates on development of CDK4/6 inhibitors. Authors have presented commentary on recent developments in the search for new selective CDK4/6 inhibitors with increased selectivity, treatment efficacy, and reduced adverse effects considering the small-molecule inhibitors and PROTACs approaches. The review focus is on SAR, selectivity against different kinases and antiproliferative activity.

Although topic is interesting, but in reviewer’s opinion the authors have not collected a unique dataset. Consequently, several closely related studies have been excluded. Generally, the manuscript is well written but not properly structured. In my opinion, the paper has some shortcomings in regard to some data collection, and reviewer is left with the feeling that collection of relevant studies has not been gathered to its full extent. I have provided my remarks on the manuscript if author considers these comments valuable to further improve the current version.

  1. The authors have stated that recent updates related to CDK4/6 inhibitors is provided. Reviewer feels that to a greater extent literature have been illustrated with examples. However, no specific time period is specified for the claimed RECENT updates in this review, and the studies have been selected randomly which has caused omission of several most relevant published data. For instance, SHR6390 (https://doi.org/10.1186/s12967-017-1231-7, doi: 1111/cas.13957), Trilaciclib (DOI: 10.1016/S1470-2045(19)30616-3, in Phase II, doi: 10.1158/1535-7163.MCT-15-0775) and BPI-16350 etc. The reviewer is left with feeling that these (as per this article's objectives) and many other findings should be included systematically to attract the readership of journal.
  2. Section 2 (Small-molecule inhibitors): In reviewer’s point of view, this section needs more attention. Section 2 summarizes major CDK4/6 inhibitors (SMIs) that have been developed or under discovery stage. A proper classification followed by a description should be given on the basis of their chemical scaffold types (classification). For instance, Pyrido[2,3-d]pyrimidin-7(8H)-one, imidazole, pyrimidine etc., and many more). In short, each class should be discussed  under separate heading (subsection).
  3. Section 3 (PROTACs): First PROTAC was discovered in 2001. To this end, few studies (10 or 12) have been included in this section for a period of almost 19 years. To this end, it is suggested that to have a more focused review article further survey should be executed for the inclusion of left-over studies.
  4. The main title should be modified and should not be misleading. To this end a specific time period should be mentioned in the title. This will benefit the readership of journal.
  5. High resolution version of Figure 3 should be incorporated.

Given these shortcomings the manuscript requires major revisions and current version does not advance for publication.‎

Author Response

We would like to express our appreciation to the reviewer for suggesting how to improve the quality of the manuscript.

  • The authors have stated that recent updates related to CDK4/6 inhibitors is provided. Reviewer feels that to a greater extent literature have been illustrated with examples. However, no specific time period is specified for the claimed RECENT updates in this review, and the studies have been selected randomly which has caused omission of several most relevant published data. For instance, SHR6390 (https://doi.org/10.1186/s12967-017-1231-7, doi: 1111/cas.13957), Trilaciclib (DOI: 10.1016/S1470-2045(19)30616-3, in Phase II, doi: 10.1158/1535-7163.MCT-15-0775) and BPI-16350 etc. The reviewer is left with feeling that these (as per this article's objectives) and many other findings should be included systematically to attract the readership of journal.

Thank you for your nice observation. In this work, the developments of CDK4/6 inhibitors during the last 5 years (from 2017 to date) have been reviewed, and this time range has been specified both in the article title (Development of CDK4/6 inhibitors: a five years update), in the abstract and in the aim of the work. Compounds indicated by the reviewer were cited in the introduction as compounds ongoing clinical trials.

  • Section 2 (Small-molecule inhibitors): In reviewer’s point of view, this section needs more attention. Section 2 summarizes major CDK4/6 inhibitors (SMIs) that have been developed or under discovery stage. A proper classification followed by a description should be given on the basis of their chemical scaffold types (classification). For instance, Pyrido[2,3-d]pyrimidin-7(8H)-one, imidazole, pyrimidine etc., and many more). In short, each class should be discussed under separate heading (subsection).

Thank you for the constructive advice. Paragraph 2 has been reorganized into subsections according to the chemical structures of the compounds.

Moreover, the research work of Abbas, S.E.-S., Synthesis and anticancer activity of some pyri-do[2,3-d]pyrimidine derivatives as apoptosis inducers and cyclin-dependent kinase inhibitors. Fut. Med. Chem. 2019, 11, 2395-2414. was added in the subsection “2.3 Pyrido-pyrimidine derivatives”.

  • Section 3 (PROTACs): First PROTAC was discovered in 2001. To this end, few studies (10 or 12) have been included in this section for a period of almost 19 years. To this end, it is suggested that to have a more focused review article further survey should be executed for the inclusion of left-over studies.

Starting from the first report of CDK4/6 PROTAC published in 2019, almost all of the published works were described. In the revised manuscript was added the paper of Steinebach, C. (Chem. Sci. 2020, 11, 3474-3486. DOI: 10.1039/d0sc00167h).

  • The main title should be modified and should not be misleading. To this end a specific time period should be mentioned in the title. This will benefit the readership of journal.

According to reviewer’s consideration, the title was changed in “Development of CDK4/6 inhibitors: a five years update”

  • High resolution version of Figure 3 should be incorporated.

Figure 3 (figure 4 in the revised manuscript) has been reorganized and the labels have been enlarged to facilitate the readers. Furthermore, the high-resolution version was incorporated.

Round 2

Reviewer 1 Report

The authors have addressed most of my suggestions.

Reviewer 2 Report

Dear Editor

To the best of my understanding, the authors have performed necessary modifications, corrections, and adequately responded to the raised concerns. I would like to recommend accepting current version of manuscript for possible publication in this Molecules journal. Please proceed as per Journal policy.

Sincerely,

Latif Mayo